# Tiger Nut (*Cyperus esculentus* L.): Nutrition, Processing, Function and Applications

**DOI:** 10.3390/foods11040601

**Published:** 2022-02-19

**Authors:** Yali Yu, Xiaoyu Lu, Tiehua Zhang, Changhui Zhao, Shiyao Guan, Yiling Pu, Feng Gao

**Affiliations:** Department of Food Science and Engineering, Jilin University, Changchun 130062, China; yuyal@jlu.edu.cn (Y.Y.); luxiaoy20@mails.jlu.edu.cn (X.L.); zhangth@jlu.edu.cn (T.Z.); czhao@jlu.edu.cn (C.Z.); shiyaog20@mails.jlu.edu.cn (S.G.); puyl20@mails.jlu.edu.cn (Y.P.)

**Keywords:** tiger nut, nutritional composition, food processing, physicochemical properties, functional characteristics, applications

## Abstract

The tiger nut is the tuber of *Cyperus esculentus* L., which is a high-quality wholesome crop that contains lipids, protein, starch, fiber, vitamins, minerals and bioactive factors. This article systematically reviewed the nutritional composition of tiger nuts; the processing methods for extracting oil, starch and other edible components; the physiochemical and functional characteristics; as well as their applications in food industry. Different extraction methods can affect functional and nutritional properties to a certain extent. At present, mechanical compression, alkaline methods and alkali extraction–acid precipitation are the most suitable methods for the production of its oil, starch and protein in the food industry, respectively. Based on traditional extraction methods, combination of innovative techniques aimed at yield and physiochemical characteristics is essential for the comprehensive utilization of nutrients. In addition, tiger nut has the radical scavenging ability, in vitro inhibition of lipid peroxidation, anti-inflammatory and anti-apoptotic effects and displays medical properties. It has been made to milk, snacks, beverages and gluten-free bread. Despite their ancient use for food and feed and the many years of intense research, tiger nuts and their components still deserve further exploitation on the functional properties, modifications and intensive processing to make them suitable for industrial production.

## 1. Introduction

The tiger nut, also known as the “underground walnut”, grows all over the world because of its high yield and broad prospects for comprehensive utilization. The tiger nut is the tiny tuber of *Cyperus esculentus* L., which can be roasted and used to be sweetmeat in Egypt [1]. Moreover, it is an important representative crop of the Spanish Mediterranean region, with an annual production of 9000 metric tons [2]. Later, it was made into a refreshing drink called “horchata de chufa” with the appearance of dairy look in the Mediterranean area, which is usually consumed in summer [1]. At present, the popularity of “horchata” has been extended from Spain to the United States, France, the United Kingdom, Portugal, China and other countries [2].

Although tiger nuts are widely cultivated around the world, the research on them is insufficient, which greatly limits their application [3]. As shown in Figure 1, it has many nutrients that can be deeply explored and contains 22.14–44.92% lipids, 3.28–8.45% proteins, 23.21–48.12% starch, 8.26–15.47% fibers and 1.60–2.60% ashes [4]. In addition, it contains bioactive substances such as organic acids, alkaloids and phenols [5]. The tiger nut is a good source of edible oils that contain a lot of monounsaturated fatty acids. The nutritional value of tiger nut oil is similar to olive oil [6]. It also contains a lot of starch—a renewable and low-cost food ingredient [7]. The content of protein is relatively small, but it is found to be suitable for diabetic patients or those with digestive dysfunctions and may prevent heart disease after consumption [3,8]. The dietary fiber in this tuber is effective in the prevention of colon cancer, obesity and gastrointestinal disorders [9]. Due to the presence of flavonoids, the tiger nut has good antioxidant properties and can be used as a source of natural antioxidants [10].

The nutritional value of tiger nuts and their products are dependent on varieties, soil conditions, growth environment, cultivation techniques and storage conditions [5,11,12,13]. Nina compared the nutritional content of black, brown and yellow oil bean varieties and found that the crude fiber of the black cultivar was higher than the brown and yellow varieties and protein content of the brown cultivar was higher than that of the black and yellow cultivar [5]. In addition, combined application of wood biochar and cow dung biochar with green manure improved soil fertility and increased growth and yield of tiger nut [11]. Tubers varied considerably among the genotypes. Refrigeration of tiger nuts resulted in higher activities of α-amylase and lipase, whilst ambient or elevated storage showed higher sugar content of tiger nuts [13]. The selection of appropriate methods are key steps in the comprehensive utilization of tiger nuts. Although tiger nut is an ancient crop, people have made some contributions in its component analysis, physicochemical properties and product development, but the breadth and depth of research are far less than that of soybean, peanut and other nut oil crops. The purpose of this article is to make this crop of interest to scientists for further research by outlining its current state of research and production. At present, the reviews of tiger nut mainly focus on recovered oil and milk processing, based on the extraction method and physicochemical properties, the analysis of nutrients is of great significance for the comprehensive utilization of tiger nut. This review article summarizes the current knowledge about the major nutrients of the tiger nut, the processing methods, their physicochemical properties, functional characteristics and the applications in the industry.

## 2. Tiger Nut Oil

The tiger nut contains a substantial amount of lipids (22.14–44.92%). The lipid profiling is similar to olive oil that is considered to be the most suitable fat for human consumption [14]. Along with the oil, it usually contains vitamin C, vitamin E, minerals potassium, calcium, magnesium and phenols [15]. The antioxidants confer to this oil higher oxidation stability compared with other vegetable oils [16]. It also contains alkaloids, saponins and tannins, which have anti-bacterial and anti-inflammatory effects [17]. Volatile compounds contribute to the sensory quality of vegetable oil. Recently, Ola et al. [18] identified a total of 75 odor-active compounds in roasted nut oil such as 5-hydroxymethylfurfural, ethyl hex decanoate and *n*-propyl-9,12-octadecadienoate, which makes the oil of a good flavor.

### 2.1. Extraction Methods of Tiger Nut Oil

Commonly, the tiger nut oil is produced by the conventional extraction method including Soxhlet extraction (SE) and mechanical expressing (ME). At present, some innovative techniques have been established for the sustainable and viable recovery of the oil, such as gas-assisted mechanical expression (GAME), mechanical compressing with enzyme assistance (MEEA), microwave-assisted extraction (MAE), microwave-ultrasonic assisted aqueous enzymatic extraction (MUAAEE), supercritical carbon dioxide fluid extraction (SC-CO_2_) and subcritical *n*-butane extraction (SBE). Details are shown in Table 1.

Among them, SE has the highest oil yield with simple operation. The oil yield by SE can reach 29.85 g/100 g dried powder. However, the time consumed and the use of organic solvent has limited its application due to safety and environmental concern [19]. ME is a relative safe oil extraction method, however, the oil yield of ME method is relatively low (19.94 g/100 g) [21]. ME of oil has cold pressing and hot pressing. The hot pressing has the advantage of higher oil yield compared with cold pressing, but has higher electric consumption. The qualities of cold-pressed oil are better, such as low acid value and good color, but the oil yield is relatively lower.

In order to increase the extraction yield without compromising the quality of the oil, some studies have implemented enzymes prior to mechanical oil extraction. Enzymes can degrade cellular wall components and thus facilitate the oil releasing from the cells. A report showed that an enzyme to substrate ratio of 1% can achieve the oil recovery rate at 20.79 g/100 g, which was higher than that of ME [22]. Additionally, an enzymatic and high-pressure pretreatment was able to increase the concentration of tocopherol and phenolic acid in the oil [24]. Bin et al. extracted tiger nut oil using microwave-assisted organic extraction (MAE), which took a shorter extraction time and had a lower solvent consumption than SE [20].

The aqua-enzymatic extraction of oil is that the cell walls of the seeds are hydrolyzed by enzymes in water as the medium. It has the advantages of environmental compatibility and simple operation [25]. However, the low oil yield and long processing time limit its application. At present, a combination of two or more technologies is the preference in the oil production [26]. Bin et al. [19] used the microwave-ultrasonic-assisted water enzymatic method (MUAAEE) to extract the oil from the tiger nuts, which greatly shortened the extraction time and improved the yield. In this method, the mechanical oscillation and stirring effect of ultrasonic waves effectively compensated for the disadvantages of uneven microwave heating. The thermal effect caused by microwave and the ultrasonic heating further improved the extraction efficiency. This method had achieved the oil recovery of 25.44 g/100 g. However, many instruments and equipment were used, and the operation was relatively cumbersome, which limited its use in industrial production.

SC-CO_2_, a novel green extraction technology, with the advantages of non-toxic, non-explosive and low-temperature [27]. SC-CO_2_ has successfully been used in the extraction of oils such as European cranberry bush, sea buckthorn, Moroccan argan and coconuts [28,29,30]. This technique highly reduces the use of toxic organic solvents. SC-CO_2_ revealed a yield of 27.79 g/100 g to extract tiger nut oil. However, it took a longer extraction time compared to other non-conventional techniques. Owing to complexity in operation and high cost of the equipment, SC-CO_2_ extraction cannot extensively be applied in practice [20]. In order to avoid long extraction time, applying SC-CO_2_ along with ME (GAME) is an alternative to improve the recovery of tiger nut oil. GAME has been previously applied to extract oil from plant matrices [31,32]. This process provided above an 80% recovery rate for tiger nut oil extraction. Furthermore, the amount of total phenolic compounds in oil after the GAME process were 2.8-folds higher than those obtained when either ME or SC-CO_2_ were used alone. Increasing the oil’s total phenolic compounds content leads to enhanced resistance to oxidation [21].

SBE is a method of extracting fat-soluble components in a closed, oxygen-free, low-pressure vessel. Compared with SC-CO_2_, SBE keeps oil quality with low pressure and temperature. Although SBE is easier for industrial production than SC-CO_2_, it still has challenges of high solvent consumption, which is not economical when directly used for raw material extraction [23].

In summary, the tiger oil can be efficiently extracted in large scale using conventional SE and ME extraction. The SE has a higher oil yield but it is easy to produce solvent residues. Tiger nut oil is produced by cold-pressing in the food industry, which has better quality, but the oil yield is relatively lower. Due to the complexity and high equipment cost, unconventional oil extraction methods are difficult to be applied to industrial production. Since a single extraction technique usually has certain shortcomings, combination of conventional with novel techniques is the direction of future development. It has been confirmed that the oil yield by GAME (28.48 g/100 g) is higher than that of single ME (19.94 g/100 g) and SC-CO_2_ (27.79 g/100 g).

### 2.2. Physiochemical and Functional Characteristics of Tiger Nut Oils

#### 2.2.1. Fatty Acid Profile of Nut Oil Produced by Different Extraction Methods

The tiger nut oil consists of predominantly long chain fatty acids (C16-C20). Among them, monounsaturated fatty acids (MUFA) account for 73.83–76.16%, polyunsaturated fatty acids (PUFA) account for 8.92–9.84% and saturated fatty acids (SFA) account for 14.60–17.12%. Oleic acid, linoleic acid and palmitic acid are the main MUFAs, PUFAs and SFAs, respectively. Oleic acid is the most predominant fatty acids in the tiger nut oil and has functional properties such as regulating blood lipids and lowering cholesterol [33]. Generally, the total fatty acid distribution of nut oil is similar to that of olive oil [2]. The MUFA and SFA content is slightly higher in nut oil than that in olive, while the PUFA content is lower. Oils with a high level of MUFA usually have great stability and health benefit properties, such as the prevention and treatment of cardiovascular disease [34].

The fatty acid composition of the tiger nut oils extracted by different methods is presented in Table 2. MAE has the highest MUFA content, which is likely attributable to the mild operation conditions. The fatty acid composition of tiger nut oil obtained by different extraction are practically similar, which indicated that fatty acids are not extracted non-selectively. This discrepancy in values is probably caused by a difference in the geographical origin, genetic history, planting environment, harvesting season and agricultural practices, since these variables can alter the lipid content of oilseeds [22].

#### 2.2.2. Chemical Characteristics of Tiger Nut Oil by Different Extraction Methods

The physicochemical characteristics of tiger nut oils obtained by different extraction methods have been listed in Table 3. The refractive index is related to the composition of the oil, reflecting the quality of the oil. Longer carbon chain and higher degree of unsaturation can result in a greater refractive index [35]. No significant differences in those oils extracted from different methods were for the refractive index. Their values were in the range of 1.46–1.48, suitable for edible oil (above 1.46). The acid value can directly reflect the amount of free fatty acids contained in the oil. The decomposition of the oil-produced fatty acids and alcohols increases the acid value. Oil extracted by ME had the lowest acid value, indicating that ME maximally keeps the quality of the oil from degradation. The high temperature and long extraction time using SE speed up the oil oxidation, leading to the highest acid value [36]. The acid value of SC-CO_2_ was high, which was probably due to the presence of carbon dioxide residues.

The peroxide value can better reflect the degree of rancidity in the initial stage of oil during oxidation. The active peroxides are produced when oil is oxidized, leading to increased peroxide values [37]. The MAE and MUAAEE showed low peroxide values, indicating that the oil obtained by those two methods contained less free fatty acids and had strong oxidative stability. It makes the deacidification in subsequent refining procedures unnecessary [20]. Moreover, microwave and ultrasonic assistance promoted the release of phenols, α-tocopherol, β-carotene and phospholipids, thereby increasing the content of unsaponifiable matter and improving the oxidative stability of oils [38]. Compared with MAE and MUAAEE, the higher temperature and longer time of SE can lead to oil peroxidation [36]. The saponification value of oil is closely related to the fatty acid molecular weight of the oil. Generally speaking, oils with a small fatty acid molecular weight have a large saponification value.

The saponification values of all oils (174.53–187.52 mg/g) were not significantly different. Iodine value is an important indicator of the degree of lipid unsaturation. According to the iodine value, the oil can be divided into non-drying oil (below100 g I_2_/100 g), semi-drying oil (100–130 g I_2_/100 g) and drying oil above (130 g I_2_/100 g). According to this standard, the tiger nut oil obtained by the above methods were all non-drying oil [20]. The high unsaturated fatty acids cause a large iodine value, which is the reason MEA has the highest value (shown in Table 2) [23,39].

All in all, mechanical pressing was used commonly for extracting the high-quality tiger nut oil with the advantage of low free fatty acids and acid value, while it has a low extraction rate compared with other methods. Microwave and ultrasound assistance can promote the release of active substances such as phenols and improve the oxidation stability of oil, which shows low peroxide value indicated in MAE (8.78 ± 0.42 meqO_2_/kg) and MUAAEE (7.63 ± 0.35 meqO_2_/kg). On the basis of traditional extraction, combination with other methods for assistance to obtain high-quality and high-yield oil may be the future development direction.

#### 2.2.3. Functional Properties of Tiger Nut Oil

In addition to the nutritional value of the tiger nut oil, the nut oil also showed biologically functional properties. In the process of oil extraction, a part of polyphenols will be extracted, which makes tiger nut oil with antioxidant properties. Moreover, high monounsaturated fatty acid (oleic acid) gives it the effect of improving blood circulation and regulating physiological metabolism [39]. Anany et al. studied the hypolipidemic effect of coconut oil blended with tiger nut oil on the rats [40]. The results showed that the mixed oil resulted in a lower serum cholesterol level of rats compared with the control. Ibitoye et al. studied the influence of tiger nut oil and soybean oil on the lipid profile of male Wistar rats [41]. They found that rats fed with tiger nut oil displayed a lower level of the total cholesterol, triglycerides, low-density lipoprotein cholesterol and higher levels in the high-density lipoprotein cholesterol and glutathione compared with that of rats fed with soybean oil. Jing et al. [42] evaluated the antioxidant activity of tiger nut oil in vitro and in vivo. Specifically, the oil at 0.8 mg mL^−1^ resulted in 26.96% DPPH^•^ radical scavenging ability. The oil at 0.2 mg mL^−1^ and 0.4 mg mL^−1^ had antioxidant abilities similar to vitamin C. Furthermore, in vivo experiments showed a maximum antioxidant effect of tiger nut oil in response to a dose of 15 mL kg^−1^ BW Day^−1^ in mice. The results clearly indicate that tiger nut oil has good antioxidant properties.

### 2.3. Applications of Tiger Nut Oil

From the point of view of commercial production, the oily bean has a high oil yield and oil quality comparable to that of olive oil and has the potential for industrialized mass production. At present, the main application of tiger nut is as a kind of edible oil [15], which is produced by cold pressing in food industry (Figure 2). Due to the ease of growth, it is widely grown and extracted oil are sold in the African market as edible oil. Moreover, Elena et al. optimized the extraction condition by manipulating alkali refining, bleaching and deodorization obtain a refined edible oil from the tiger nuts [43].

Tiger nut oil can replace animal fat in meat products [44]. It was applied as a substitute for animal fat in the production of deer and beef burgers. By the addition of 3 g/100 g of tiger nut oil, deer burgers had a similar textural and sensory properties with the burgers containing pork fat as a lipid source [45]. The total replacement of beef fat by tiger nut oil emulsion in beef burgers resulted in a healthier meat product with rich unsaturated fatty acids but low total fat and saturated fatty acid contents, which improved the nutritional quality of burgers [46]. The researchers observed a significant decrease in the fat content in burgers with animal fat replacement. The reformulated batches could be labelled as “low fat” burgers, which is a great advantage for high consumer acceptability.

## 3. Tiger Nut Starch

Starch, the main reserve substance in plants, is a rich, non-toxic, renewable and low-cost food ingredient. It accounts for about 80–90% of all polysaccharides in human food [47]. Tiger nut, a plant, is considered to be suitable for starch production under the industrial mode and has the advantages of high yield, easy cultivation and especially easy extraction for starch [7]. Tiger nut starch has similar characteristics to corn starch, potato starch and sweet potato starch, such as an oval particle shape, hygroscopicity and adhesiveness [48,49].

### 3.1. Extraction and Physicochemical Properties of Tiger Nut Starch

At present, the starch from the tiger nut is produced by extraction methods such as the alkaline method [50,51], sodium metabisulfite extraction [52,53] and water extraction [54]. The detailed extraction process is summarized in Table 4.

The relative proportion of amylose and amylopectin is largely responsible for the functional and physicochemical properties of starches, such as gelatinization, paste viscosity, gel stability and solubility. Among them, the starch extracted by the water extraction has a slightly higher amylose content, which enhances the film-forming ability. The starch obtained by sodium metabisulfite extraction has a higher amylopectin content, which exhibits a pasting property. Compared with potato, cassava and other native starches (15–30%), tiger nut starch has a lower amylose content but higher amylopectin content, indicative of stronger associative forces suitable to be applied as a binder.

The physicochemical properties of tiger nut starch are closely dependent on the seed varieties (black and yellow) and the pretreatment of the nut for oil. The study showed wide variations in some physicochemical properties of starches isolated from the two types of tubers (black and yellow) [54]. Both starches largely consist of spherical granules with smooth surfaces, but the black nut starch contained smaller granules [54]. Starch gels from the yellow variety were clearer, softer and more adhesive, cohesive and stable after freeze-thaws. The black nut starch showed higher peak viscosity and setback viscosity and lower breakdown, which made it suitable to be used as food packaging materials and drug transport films. The structural and functional properties of tiger nut starches were different with nut meals after hexane extraction, hot pressing, cold pressing and subcritical fluid extraction. The ordered structure of starch granules was caused by the various oil extraction processes, which led to higher solubility and swelling power, lower freeze-thawing stability of starch compared to untreated nuts [58].

No matter which method is used, the tiger nut starch obtained has common properties. Tiger nut starch is a brilliant white, odorless powder with a warm bland taste and smooth texture. It consists of elliptical to spherical and small- to medium-sized granules with particle sizes ranging from 2 to 17μm [53]. It has a relatively high content of amylopectin with good thickening and bonding effects.

### 3.2. Applications of Tiger Nut Starch

Tiger nut starch has similar viscosity characteristics to native starch, but its gel texture properties (hardness, elasticity, cohesion and chewiness) are higher than those of traditional corn and sweet potato starch. In addition, its freeze-thaw stability is better than that of corn starch, which can be used in cold drinks and frozen foods and has a wide range of application values. Therefore, it has potential commercial value and has been used in the food and pharmaceutical industry.

The tiger nut starch gels are clear, soft, adhesive and cohesive, which is a good choice for use as a thickener, binder and humectant in food products, such as soups and dessert powders [54]. In addition, it has a high gel strength and thawing stability in the hydrated state, which is suitable for the production of jelly, cold drinks and candies [50]. Starch is one of the most frequently used excipients in pharmaceutical formulations and is included in the GRAS (Generally Recognized as Safe) list of the World Health Organization. The tiger nut starch shows high compatibility and binding ability, has good swelling power, density water absorption and good binding efficiency with medicines, which conforms to the pharmaceutical standard [48]. So, the nut starch can be used as an excipient, binder, filler and lubricant for the production of pharmaceutical preparations [53]. Indeed, this starch showed excellent binder activity, mechanical strength, fluidity and release properties when it was used as glidant and binder in the production of metronidazole and ascorbic acid tablets [52,59]. The native tiger nut starch showed flow properties comparable to maize and potato starch, as well as excellent binder activity when metronidazole tablets’ hardness and friability were evaluated [48].

Modified starch overcomes limitations of natural starch, thereby improving its practicality in industrial applications. A low digestible and viscous starch was obtained by modifying tiger nut starch used the pullulans hydrolysis, which is highly valued as a component in functional foods such as improving lipid and cholesterol metabolism and strengthening intestinal function [50]. Furthermore, modified starch was used in the productions of ascorbic acid granules and ibuprofen tablets in the pharmaceutical industry to improve the tablet’s mechanical strength, swelling capacity, adhesion and release properties [48,60].

## 4. Tiger Nut Protein

After oil extraction, the meals as the by-product of oil extraction are currently used for animal feed or directly discarded, causing a lot of waste and environmental pollution. The tiger nut protein is suitable for patients with diabetes or digestion-dysfunction diseases and plays an important role in the prevention of heart disease [3]. Therefore, isolated protein from the meals after the oil extraction of tiger nuts can be further processed into health products.

### 4.1. Extraction of Tiger Nut Protein

Nowadays, shown in Table 4, the main extraction methods of tiger nut protein are the alkali extraction–acid precipitation [55], ammonium precipitation [56] and reverse micelle extraction [57].

For tiger nut protein extraction, traditional alkaline extraction–acid precipitation was widely used with the advantages of simple operation and low cost. However, the protein biological activity was compromised during processing, and the discharged acid and alkali may impact on the environment. Compared with the alkaline solution and acid precipitation method, the protein obtained by ammonium precipitation has higher purity (83.4%) and quality, as the structural properties of protein are mildly affected. However, this method is time-consuming and complicated to operate and not suitable for industrial production. The reverse micelle technology is a new effective separation technology. It uses surfactants to dissolve in a non-polar solution to form a polar core, which can solubilize the protein, thereby isolating the protein from the solvent and reducing the denaturation of the protein by the solvent [61]. At present, the molecular mechanism of reverse micelle extraction of proteins is not yet clear, and the use of organic solvents will bring safety hazards which is not suitable for large-scale industrial production.

### 4.2. Composition and Nutritional Evaluation of Tiger Nut Protein

The molecular weight of the tiger nut protein is 5.5–88 kDa. The composition of tiger nut protein is 47.5% gluten, 31.8% albumin, 4.7% globulin and 3.8% prolamin [59]. Furthermore, other research has shown different composition results with the highest gluten content (about 82–91%) in tiger nut protein [62]. The reason for this difference was possibly because that the proximate composition of tiger nut was mainly dependable on their geographical origin [12]. The various components of tiger nut protein have different effects on their functional properties, and it is meaningful to find protein components suitable for the production of different products for the comprehensive utilization of tiger nut protein.

Plant proteins can now be regarded as functional ingredients or as biologically active components more than as essential nutrients. Because functional properties of protein often have a tight relationship with its amino acid profile, so the amino acid composition represents the potential quality of a plant protein. Researchers evaluated the nutritional value of tiger nut protein by considering both the essential amino acid profile and protein digestibility. Tiger nut protein contains 18 kinds of amino acid, of which the essential amino acids accounts for 46.03%, largely exceeding the value specified in the WHO/FAO model (36%), higher than soy protein (41.3%) and close to egg protein (48.8%). According to the literature, lysine is the first limiting amino acid of some nuts (Brazil nuts, macadamia nuts and almonds and some cereal grains (rice, white flour, corn, etc.)). However, the amounts of lysine in tiger nut are much greater, accounting for 15.4% of essential amino acids, which is higher than 13.3% of soybean [63]. Therefore, with respect to lysine amino acid contents, cereal and tiger nut proteins are nutritionally complementary. The first limiting amino acids of the tiger nut protein is methionine. Taking the score of the first limiting amino acid as the protein amino acid score, the amino acid score of tiger nut protein is 78.9, which is higher than soy protein (51.4) and lower than egg protein (97.2). The digestibility of the tiger nut protein was also evaluated. The results showed that tiger nut protein has 76% of in vitro digestibility, slightly lower than soy protein (86%) [63]. In conclusion, the protein derived from tiger nut is considered high quality and commercial value based on amino acid profile and bioavailability. On the other hand, the protein of tiger nut is limited in methionine. We recommend the tiger nut as the sources of complementary protein.

### 4.3. Physiochemical Characteristics of Tiger Nut Protein

The tiger nut protein extracted by alkali extraction–acid precipitation with the solubility of approximately 41.29%, emulsifying activity index of 42% and emulsion stability index of 12.6% [58], which are lower than soybean, peanut and corn proteins. The functional properties of natural tiger nut protein cannot meet the requirements for food processing and production; therefore, it is necessary to improve the physical and chemical properties of the protein by biochemical modification.

The effective modification methods such as ultrasound technology and pH shift have been used in the modification of tiger nut protein obtained by alkali extraction–acid precipitation extraction. The increase in ultrasound power and pH value induced the increase in the solubility and emulsifying properties, respectively. Ultrasound treatment can cause a cavitation effect, which destroys the insoluble protein aggregates, reduces the particle size and enhances the expansion of the hydrophobic groups of the protein molecules [58]. These treatments directly lead to an increase in the solubility and emulsifying properties of the nut protein. With the increasing pH, the surface hydrophobicity and emulsification ability of the protein firstly decreased and then increased. In the proper pH range, the compact protein structure unfolds, the hydrophobic group wrapped in the protein molecule is exposed and, thus, the surface activity is enhanced, leading to an increase in emulsification [64]. Consequently, the ultrasound and pH treatment could be applied to make the tiger nut protein suitable for different food processing conditions in the actual production.

### 4.4. Applications of Tiger Nut Protein

Natural antioxidant peptides have the characteristics of low toxicity, high efficiency and easy absorption, which are becoming more and more popular for their use as an antioxidant. Yin et al. analyzed the antioxidant capacity of Tiger nut antioxidant peptide prepared from Tiger nut protein extracted by alkali extraction–acid precipitation. The studies showed that using alkaline protease to prepare antioxidant peptides with the ACE inhibition rate of 74.16% [65].

There are relatively few studies and applications on tiger nut protein, the future development should focus on the following technologies: functional and nutritional properties of each component, application more than peptide in the food industry.

In addition to the macromolecular nutrients such as oil, starch and protein, studies have shown that there are small molecules or secondary metabolites that are mainly responsible for biological activity, including flavonoids, minerals, vitamins and Stigmasterol [17]. Among all of the secondary metabolites, the flavonoids quercetin and myricetin show a wide range of biological activities that include strong anti-oxidant, anticarcinogenic, anti-inflammatory and antidiabetic effects. Due to the presence of quercetin, this tuber has aphrodisiac activity, enhancing male sexual libido and performance [17]. The minerals in the tiger nut are sodium, potassium, calcium, iron, magnesium, zinc, copper and phosphorus. The high potassium (110.70–21.95 mg/100 g) to low sodium ration (99.95–105.6 mg/100 g) of tiger nuts may be imperative in diet formulations for patients with high blood pressure and edema [5]. Moreover, the presence of vitamin C and E gives the tuber the function of preventing scurvy and promoting liver detoxification [17]. The cholesterol-lowering activity and the protective effect against cardiovascular disease of the tuber may be due to the stigmasterol. However, the isolation, identification and quantification of these active ingredients, which have not been adequately studied, are necessary for the development of functional properties of tiger nut.

## 5. Tiger Nut-Based Food Products

The tubers are rich in nutrients and have a wide range of applications. They can be made into snacks, beverages [2] and gluten-free [66] bread due to the flavor of sweet and nutty, dietary fiber and fermentable sugars. In addition, it has certain medical properties. The main product application of tiger nut has been shown in Figure 3, and the industrial products have been shown in Figure 4.

### 5.1. Tiger Nut Milk

“Horchata de chufa” is a plant milk produced by tiger nut and is still popularly consumed in Spain. Compared with animal milk, plant milk has many positive health effects on the human body, especially for people with milk allergies and lactose intolerance. The tiger nut milk is produced by the steps of wet milling, filtration, addition of ingredients, sterilization, homogenization, aseptic packaging and refrigeration. It contains phenolic compounds, unsaturated fatty acids and biologically active substances [62].

In spite of the advantages of “Horchata de chufa”, it has not been widely distributed worldwide due to the high microbiological loads of harvested tubers and short shelf life. Moreover, conventional thermal treatments such as pasteurization and sterilization result in an undesirable loss of the most appreciated sensory characteristics of “Horchata de chufa”. Owing to this, the food industry looks for alternative technologies that can improve the microbiological quality of “Horchata de chufa” while preserving the sensory characteristics.

Tiger nut contains a high amount of starch that is easy to gelatinize during the pasteurization process, leading to the coagulation of the milk and limits the output. Exogenous amylase can be used to increase the yield through the in situ hydrolysis of starch, which is a process that increases the sweetness of milk and is suitable for the industrial production of tiger nut milk [67]. In addition, heat treatment could cause the loss of nutrients such as the total protein, phenols and vitamins of the tiger nut milk. Compared with thermal processing, non-thermal processing retains the nutritional quality and is more suitable for the production of tiger nut milk [68]. In non-thermal processing, ultra-high-pressure homogenization [69], short-wave ultraviolet treatments [70] and pulsed electric field [71] can effectively inhibit the growth of microorganisms and extend the shelf life. On the other hand, studies have shown that the microencapsulation of tiger nut milk by a blend of inulin and modified tiger nut starch can make a product with good characteristics and shelf life [72].

### 5.2. Comprehensive Utilization of Tiger Nut Milk By-Products

The by-products during tiger nut milk production account for about 60% of the harvested material and can be used as sources of polysaccharides, fibers, oil and antioxidants (vitamin E and polyphenols) [6]. The by-products are divided into two parts by pressing and filtering (shown in Table 5): the liquid phase contains biologically active substances such as phenols [73] and the solid phase include dietary fibers [74].

The liquid phase of milk by-products is an important source of natural antioxidants, which have antioxidant properties and can inhibit lipid peroxidation. Elena et al. [75] extracted phenolic compounds from the by-products, and the results showed that 222.6 mg of gallic acid equivalents was obtained from 100 g of matter. Therefore, it can be regarded as a valuable source of phenolic compounds with potential applications in food industry, nutraceuticals and cosmetics. The liquid by-products can be used as an ingredient replacing water in a cooked pork liver meat product. The pig liver paste produced with 50 and 100% of water replacement had a higher content of heme and a low degree of formation of myoglobin, which had good sensory properties [76]. Sanchez et al. proposed the liquid by-products of the tiger nut milk as carbon source for growth of probiotic bacteria (*Lactobacillus acidophilus* and *Bifidobacterium*) and metabolic activity. The results indicated that it has a high potential for probiotic microorganisms’ growth and stability as fermentable substrates [77], but more studies are required to establish the doses, the stability and its compatibility with other probiotic microorganisms.

The solid by-products of tiger nut milk are the main source of fiber. Compared with other sources of dietary fibers (oat bran, rice bran, sugarcane waste), tiger nut fiber has higher water holding capacity, oil holding capacity and emulsification stability and low water absorption [78]. Studies have shown that foods rich in dietary fiber are essential for preventing colon cancer, constipation, obesity and cardiovascular disease [9]. Adding solid co-products to cereal foods (such as chips, breakfast cereal or dry pasta) would reduce the capacity of generic vitreous wheat-based matrix to interact with solvents from the 10% substitution level, with diminished surface tension, wettability and water–oil diffusion properties [79]. Tiger nut fibers have the ability to lower cholesterol, which can be used in pork pies [80], pork burgers and dry-cured pork sausages [81] with higher nutritional quality (lesser percentage of fat and more total dietary fiber content) and cooking characteristics. Furthermore, the solid by-products of tiger nut milk can also be used as a carrier of unsaturated fatty acids oil rich in dry-cured sausages for avoiding the problems related with its incorporation in the meat matrix and for controlling lipid oxidation [82].

### 5.3. Gluten-Free Bakery Products

Demand for gluten-free products is increasing as a result of the increase in the prevalence of celiac disease, but high-quality gluten-free bread production is a big challenge [83]. To improve the organoleptic properties and shelf life, cereal flours, starches, proteins, hydrocolloids and emulsifiers are usually mixed in the production of gluten-free bread. The tiger nut was evaluated as a new ingredient in gluten-free products [83]. In gluten-free bread, tuber flour can be used as emulsifier and shortening when combined with chickpea flour; as a result, the bread products have good baking characteristics in color, hardness and volume [84]. Moreover, tiger nut flour can be added as a functional ingredient to promote a reduction in diameter, expansion ratio, true density and total pore volume in the extrudates and the ash, protein and total phenol content of extruded snacks [85]. In a subsequent study, compared with biscuits containing only corn flour, biscuits made with tiger nut powder have better shape, cross-sectional area, hardness and surface appearance [86].

### 5.4. Tiger Nut Beverage

Due to the good sensory properties such as sweet taste, it has a wide range of applications in beverage. Badejo et al. [87] used tiger nut as a synergistic vehicle in combining leafy vegetable (*Momordica charantia* and *Vernonia amygdalina*) of functional beverages. The addition of tiger nut reduced bitter taste and increased antioxidant and anti-diabetic potentials of the beverage. Furthermore, carbonated tiger nut beverages mixed with apple, pineapple and coconut showed good qualities in chemical composition, physiochemical properties, microbiological evaluation, vitamins, mineral content and shelf life, which are all in compliance with standard specifications [88].

### 5.5. Tiger Nut Pasta

There has been a trend of using non-durum wheat ingredients to enhance the nutritional and functional characteristics of pasta. Tiger nut was mixed with wheat flour to produce composite fresh pasta. The addition of tiger nut powder significantly improved the fiber, fat and mineral qualities of the product. The substitution of 30% tiger nut powder can ensure the fiber content of products exceeds 3%. However, the gluten protein structure needs to be strengthened in order to reduce cooking loss and increase the firmness of the pasta [89]. The powder (20 and 40%) and xanthan gum (1%) incorporation into fresh egg pasta compensates for the problem displaying a good gluten network. It has shown that the rheological and structure properties of pasta were modified significantly with the optimum cooking time for 2 min [90]. Uncooked pasta has good water absorption and swelling index, less cooking loss. The cooked pasta has good hardness, elasticity, color and sensory attributes [91].

## 6. Medicinal Properties of Tiger Nut

Tiger nut has been shown to be a good source of bioactive substances. It contains polyphenols, flavones, minerals, essential fatty acids and vitamins C, D and E, etc. Therefore, it might possess medicinal properties. Ademosun & Oboh sought to investigate the radical scavenging ability and in vitro inhibition of lipid peroxidation, α-amylase and α-glucosidase activities of tiger nut extracts. The assessed antioxidants of aqueous extracts of the tiger nut as typified by 1,1-diphenyl-2 picrylhydrazyl (DPPH) and hydroxyl (OH) radicals showed scavenging abilities and the inhibition of Fe^2+^-induced malondialdehyde (MDA) production in rat pancreas in vitro. The results showed that the EC50 of DPPH· and OH· scavenging abilities, Fe^2+^-chelating ability, inhibition of Fe^2+^-induced MDA production and inhibition of α-amylase and α-glucosidase activities by aqueous extracts of the tiger nut are 9.63 ± 0.7 mg/mL, 3.01 ± 0.12 mg/mL, 0.72 ± 0.07 mg/mL, 2.09 ± 0.10 mg/mL and 0.76 ± 0.06 mg/mL, respectively [92]. The findings support the hypothesis that tiger nut maybe beneficial in the management of type 2 diabetes. In other studies, the tiger nut powder was administered to the rats daily showed anti-inflammatory and anti-apoptotic effects to prevent testicular dysfunction [93], ameliorated male arousal [94], reduced diarrheal symptoms in albino rats [95] and reduced oxidative stress in liver and inflammatory with atherosclerosis [96]. The above results are most likely related to the presence of alkaloids, quercetin, vitamins, steroids and zinc, etc., in tiger nuts. Therefore, in addition to be the food and industrial materials, tiger nuts may also be developed into functional foods. Nevertheless, future investigations are warranted to confirm these effects of tiger nut in humans and to identify the specific ingredients that play the medicinal role.

## 7. Conclusions

Tiger nut is a valuable source for diverse nutrients, such as oil, starch, fiber, protein, phenolic compounds, etc. Several conventional (Soxhlet extraction and mechanical expressing) and alternative innovative methods (gas, enzyme, microwave and ultrasonic assistance, SC-CO_2_ and SBE) have been developed for the efficient recovery of tiger nut oil. GAME has a high oil yield, similar to SE. ME has the advantages of low free fatty acid and acid value, while having a lower yield. Tiger nut starches obtained by alkaline, sodium metabisulfite and water extractions have relatively high contents of amylopectin with good thickening and bonding effects. The recovery of protein from tiger nut is performed using the alkali extraction–acid precipitation, ammonium precipitation and reverse micelle extraction methods. The protein obtained by ammonium precipitation has higher purity and quality, but the process is more time-consuming. The combination of multiple techniques is the direction of the future development of tiger nut. In addition, it can be used in production of edible oil, jelly, candy, pharmaceutical standard, antioxidant peptide, etc., owing to diverse nutrients. On the other hand, it could be developed with whole tuber in the production of milk, gluten-free bakery, beverages and pasta. Innovative methods are still needed to fully utilize the whole tiger nut for more healthy products.

Tiger nut is an excellent and healthy crop from both a nutritional and commercial point of view. Green production and comprehensive utilization are the future development direction. In the process of the full utilization of tiger nut, the first step is to extract oil, and the extraction method of oil has a certain influence on the yield and physiochemical properties of subsequent starch, protein and fiber. Therefore, the use of innovative combination methods to fractionate and extract the nutrients oil, starch, protein and fiber in a tuber is worth thinking about in the future. It is meaningful to make modifications to the natural tiger nut starch and develop new varieties of starch, making it adaptable to different processing conditions and use it in biological preservative films. In addition, investigating the functional properties of protein and using it in baking and nutritional meal replacement foods is crucial to the study of tiger nut protein. Due to its high fiber content, it can be used in the development of food for improving gut health and weight loss.

## Figures and Tables

**Figure 1 foods-11-00601-f001:**
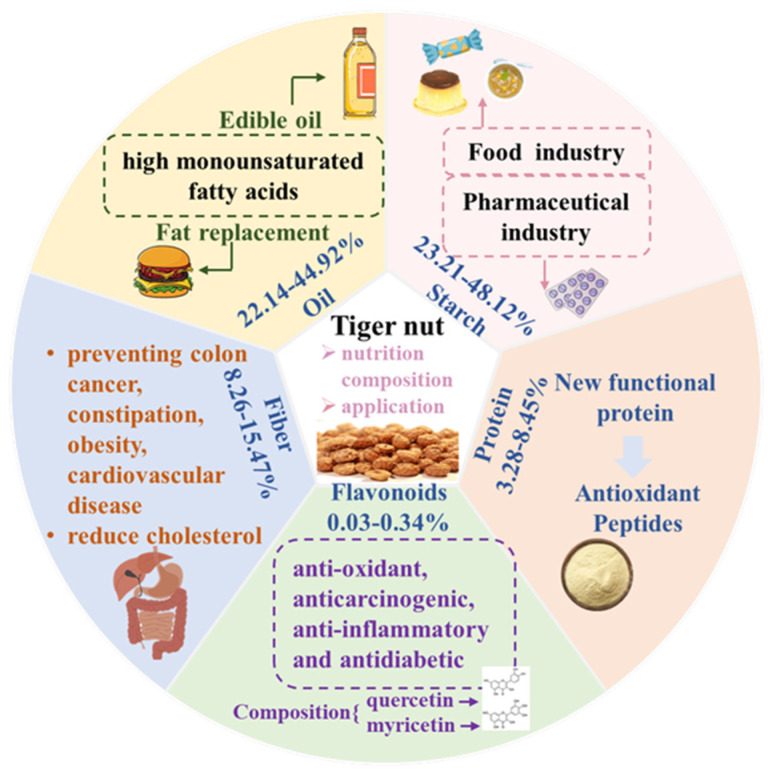
The main nutrition composition and nutrient-based applications of tiger nut.

**Figure 2 foods-11-00601-f002:**
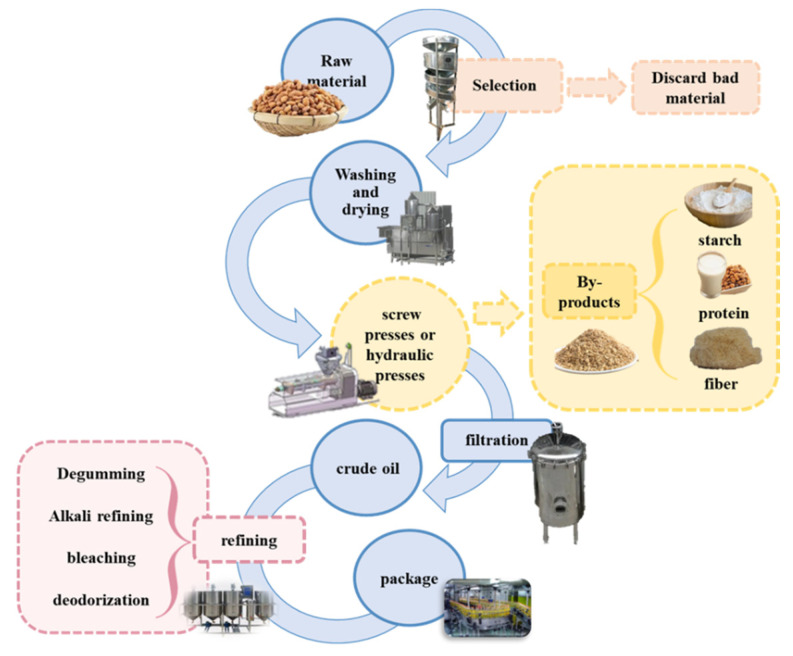
The preparation of tiger nut oil in food industry and by-products generated.

**Figure 3 foods-11-00601-f003:**
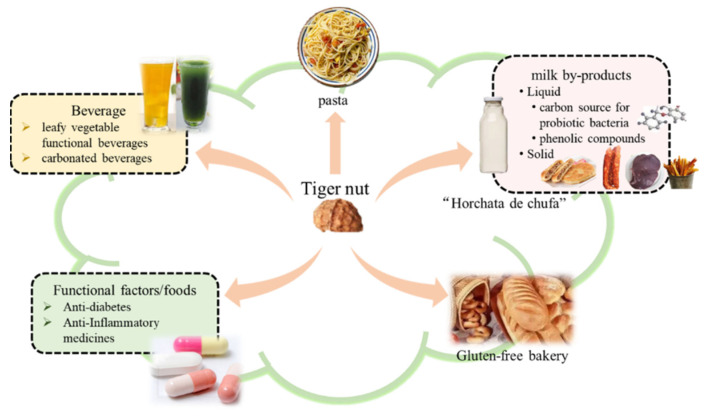
Applications of the whole tuber of the tiger nut in the food industry.

**Figure 4 foods-11-00601-f004:**
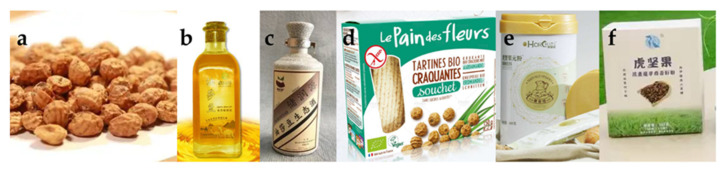
Tiger nut tuber and its products ((**a**) tuber; (**b**) edible oil; (**c**) liquor; (**d**) crunchy bio toast; (**e**) solid powder drink; (**f**) meal replacement powder).

**Table 1 foods-11-00601-t001:** Extraction technologies of tiger nut oil.

Method	Condition	Oil Yield	Reference
SE	Powder: n-hexane = 1:10(*w*/*v*), extraction temperature 80 °C, 6 h	29.85 g/100 g	[19]
MAE	petroleum ether and acetone (2:1, *v*/*v*), microwave power 420 W, liquid to solid ratio 7.0 mL/g, 75 °C and 55 min	24.12 g/100 g	[20]
ME	Pressing temperature 40 °C, 120 min, 30 MPa, speed 0.1 mm/s	19.94 g/100 g	[21]
GAME	Temperature 40 °C, CO_2_ pressure 20 MPa, CO_2_ flow 8.5 kg/h, pump pressure 30 MPa, 120 min	28.48 g/100 g	[21]
MEEA	Protease, α-amylase and Viscozyme L(1/1/1, *w*/*w*/*w*), mixed enzyme addition 1%, pH 8, 40 °C	20.79 g/100 g	[22]
MUAAEE	Cellulase, pectinase and hemicellulase (1/1/1, *w*/*w*/*w*), 40 °C, pH 3.5, ultrasonic power 300 w, microwave power 300 w, radiation time 30 min, enzyme concentration 2%, liquid to solid ratio 10 mL/g and enzymolysis time 180 min	25.44 g/100 g	[19]
SC-CO_2_	Extraction temperature 60 °C, pressure 28 MPa, 90 min	27.79 g/100 g	[20]
SBE	Extraction temperature 40 °C, extraction time 50 min	26.03 g/100 g	[23]

SE: Soxhlet extraction; ME: mechanical expressing; GAME: gas assisted mechanical expression; MEEA: mechanical compressing with enzyme assistance; MAE: microwave-assisted extraction; MUAAEE: microwave-ultrasonic assisted aqueous enzymatic extraction; SC-CO_2_: supercritical carbon dioxide fluid extraction; SBE: subcritical n-butane extraction.

**Table 2 foods-11-00601-t002:** Fatty acid composition of tiger nut oil under different extraction methods.

Fatty Acid	SE	ME	MAE	MUAAEE	SC-CO_2_	SBE
Palmitic acid (C16:0)	11.65 ± 0.28 ^c^	11.98 ± 0.00 ^b^	12.08 ± 0.16 ^ab^	11.86 ± 0.23 ^bc^	12.36 ± 0.01 ^a^	12.14 ± 0.01 ^ab^
Palmitoletic Acid (C16:1)	0.26 ± 0.00 ^a^	- *	0.22 ± 0.00 ^b^	0.25 ± 0.00 ^a^	- *	- *
Cydonic acid (C17:0)	0.07 ± 0.00 ^a^	- *	0.06 ± 0.00 ^a^	0.07 ± 0.00 ^a^	- *	- *
Stearic acid (C18:0)	2.43 ± 0.05 ^d^	4.92 ± 0.00 ^a^	2.24 ± 0.04 ^e^	2.37 ± 0.07 ^d^	4.76 ± 0.00 ^c^	4.84 ± 0.00 ^b^
Oleic acid (C18:1)	74.52 ± 0.51 ^bc^	73.97 ± 0.01 ^c^	75.60 ± 0.58 ^a^	74.73 ± 0.46 ^b^	73.83 ± 0.01 ^c^	74.10 ± 0.01 ^bc^
Linoleic acid (C18:2)	9.63 ± 0.12 ^a^	8.75 ± 0.00 ^b^	8.85 ± 0.13 ^b^	9.46 ± 0.18 ^a^	8.86 ± 0.00 ^b^	8.75 ± 0.00 ^b^
α-linolenic acid (C18:3)	0.21 ± 0.00 ^c^	0.38 ± 0.00 ^b^	0.91 ± 0.00 ^a^	0.2 ± 0.00 ^d^	0.19 ± 0.00 ^e^	0.17 ± 0.00 ^f^
Arachidic acid (C20:0)	0.45 ± 0.00 ^a^	- *	0.41 ± 0.00 ^c^	0.42 ± 0.00 ^b^	- *	- *
Eicosenoic acid (C20:1)	0.28 ± 0.00 ^a^	- *	0.28 ± 0.00 ^a^	0.29 ± 0.00 ^a^	- *	- *
Saturated fatty acids	14.25 ± 0.05 ^e^	16.90 ± 0.01 ^c^	14.69 ± 0.05 ^d^	14.65 ± 0.07 ^d^	17.12 ± 0.01 ^a^	16.98 ± 0.01 ^b^
Monounsaturated fatty acids	75.06 ± 0.12 ^c^	73.97 ± 0.01 ^de^	76.16 ± 0.12 ^a^	75.24 ± 0.13 ^b^	73.83 ± 0.01 ^e^	74.10 ± 0.01 ^d^
Polyunsaturated fatty acids	9.84 ± 0.07 ^a^	9.14 ± 0.00 ^c^	9.04 ± 0.07 ^c^	9.72 ± 0.09 ^b^	9.05 ± 0.00 ^c^	8.92 ± 0.00 ^d^
Reference	[19]	[32]	[23]	[19]	[32]	[32]

* Not detected. Different letters in a row indicate significant differences at the 5% level. SE: Soxhlet extraction; ME: Mechanical expressing; GAME: Gas-assisted mechanical expression; MEEA: Mechanical compressing with enzyme assistance; MAE: Microwave-assisted extraction; MUAAEE: Microwave-ultrasonic-assisted aqueous enzymatic extraction; SC-CO_2_: Supercritical carbon dioxide fluid extraction; SBE: Subcritical n-butane extraction.

**Table 3 foods-11-00601-t003:** Physicochemical characteristics of tiger nut oil under different extraction methods.

Method	Refractive Index (25 °C)	Acid Value (mg/g)	Peroxide Value (meqO_2_/kg)	Saponification (mg/g)	Iodine Value (g/100 g)	Reference
SE	1.46 ± 0.00 ^a^	4.15 ± 0.32 ^a^	16.26 ± 0.53 ^b^	187.25 ± 1.42 ^a^	84.78 ± 1.13 ^ab^	[19]
ME	1.48 ± 0.00 ^b^	1.90 ± 0.01 ^e^	15.76 ± 0.00 ^b^	174.53 ± 0.62 ^b^	67.35 ± 0.49 ^c^	[32]
MAE	1.46 ± 0.00 ^b^	2.26 ± 0.17 ^d^	8.78 ± 0.42 ^c^	185.67 ± 1.37 ^a^	85.41 ± 1.06 ^a^	[23]
MUAAEE	1.46 ± 0.00 ^b^	2.35 ± 0.13 ^d^	7.63 ± 0.35 ^d^	187.52 ± 1.23 ^a^	83.67 ± 0.85 ^b^	[19]
SC-CO_2_	1.46 ± 0.00 ^b^	3.39 ± 0.16 ^b^	15.76 ± 0.00 ^b^	175.33 ± 1.61 ^b^	65.60 ± 0.14 ^d^	[32]
SBE	1.46 ±0.00 ^b^	2.86 ± 0.02 ^c^	23.64 ± 0.00 ^a^	176.71 ± 0.81 ^b^	66.15 ± 0.64 ^cd^	[32]

Different letters in a row indicate significant differences at the 5% level. SE: Soxhlet extraction; ME: Mechanical expressing; GAME: Gas-assisted mechanical expression; MEEA: Mechanical compressing with enzyme assistance; MAE: Microwave-assisted extraction; MUAAEE: Microwave-ultrasonic-assisted aqueous enzymatic extraction; SC-CO2: Supercritical carbon dioxide fluid extraction; SBE: Subcritical n-butane extraction.

**Table 4 foods-11-00601-t004:** Extraction technologies of tiger nut starch and protein.

Method	Condition	Amylose (g/g)	Amylopectin (g/g)	Protein Content	Reference
Alkaline method	solid–liquid ratio of 1:15, 0.15% sodium hydroxide, soak 30–40 min, 45 °C drying 24 h	16.18	83.82	-	[51]
Sodium metabisulfite extraction	sodium metabisulfite solution (0.075% *w*/*v*) dispersing, 150 µm muslin cloth filtering, air dried for 10 h, further dried for 5 h at 50 °C	11.5	88.5	-	[53]
Water extraction	milled with water, Gauze filter, 150 μ mesh filter, 40 °C drying 18 h	19.1	80.9	-	[54]
Alkali extraction-acid precipitation	alkali extraction pH 8.5, extraction time 1.5 h, acid precipitation pH 4.5, incubated at 4 °C for 2 h	-	-	72%	[55]
Ammonium precipitation	pressing of the pre-soaked mush, filter through a 13-µm pore-size membrane, 50% *w*/*v* ammonium sulphate, 16 h, 6- to 8- kDa cut-off membrane tube dialyzing	-	-	83.5%	[56]
Docusate sodium/isooctane reverse micelle extraction	The pre-extraction condition are as follows: 0.05 g/mL feed volume, 40 °C, 86 min, pH 7, the docusate sodium mass concentration is 0.12 g/mL, the water content is 16, the KCl concentration is 0.02 mol/L. The back-extraction conditions were as follows: stripping time was 5 min, the pH of the aqueous phase was 11, and the KCl concentration was 1.2 mol/L.	-	-	-	[57]

The content of amylose and amylopectin was determined by colorimetric method based on amylase–iodine complex formation. The respective protein content in table was determined using the Kjeldahl method (Nitrogen content to protein conversion factor was 6.25).

**Table 5 foods-11-00601-t005:** Application of tiger nut milk by-products in the food industry.

Tiger Nut By-Products	Source	Food Product
liquid phase	phenolic compounds	natural antioxidants
	replacing water	cooked pork liver meat product
	carbon source for growth of probiotic bacteria	fermentable substrates
solid phase	Tiger nut fiber	cereal foods (such as chips, breakfast cereal or dry pasta)
		pork pies, pork burgers and dry-cured sausages

## Data Availability

Not applicable.

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
