# Peer review of "Tiger Nut (*Cyperus esculentus* L.): Nutrition, Processing, Function and Applications"

_foods, 2022, doi:10.3390/foods11040601_

Round 1

Reviewer 1 Report

I reviewed the manuscript entitled, Tiger nuts (Cyperus esculentus L.): nutrition, processing, function and applications. The review is well written; however, several revisions are required.

Title: replace Tiger nuts with Tiger nut

Abstract: what are the conclusions of the study? Authors must discuss the findings of the review and future recommendation in abstract. The present form is almost meaningless

The entire format is not according to the journal style. Authors must need to read journal guidelines and follow the journal format.

Figure 1. increase the font size of descriptors

Section 4.2. Composition and Nutritional Evaluation of Tiger Nut Protein: Please provide detailed discussion

Section 4.3. Activity of Tiger Nut Protein, which activity?

5.2. Comprehensive Utilization of Tiger Nut By-Products

Please provide in Table form for the tiger nut by-products, source and food product

  1. Medicinal Properties of Tiger Nut should be more elaborated. Is it a whole tiger nut or including by-products of it? Detailed discussion should be necessary

References: none of the references are according to journal format. It seems possible that the authors are no aware of Foods guidelines

Scientific names should be in Italics and journal names should be in Italics (according to food references style). Authors should check all the references and make sure to follow journal guidelines.

Find the attached document for other suggestions 

Author Response

We want to express our sincere gratitude to the reviewer for your effort and patience in reviewing our manuscript. We appreciate for Reviewer’s warm work earnestly. We addressed all the points as summarized below. All the changes were highlighted in red.

Point 1: Title: replace Tiger nuts with Tiger nut

Response 1: Thanks for your comment. We revised the title to “Tiger nut (Cyperus esculentus L.): nutrition, processing, function and applications”

Point 2: Abstract: what are the conclusions of the study? Authors must discuss the findings of the review and future recommendation in abstract. The present form is almost meaningless

Response 2: Thanks for your comment. We are glad to accept your suggestion. And the abstract section has been revised.

Page 1, line 10:

The tiger nut is the tuber of Cyperus esculentus L. which is a high-quality wholesome crop that contains lipids, protein, starch, fiber, vitamins, minerals and bioactive factors. This article systematically reviewed the nutritional composition of tiger nuts, the processing methods for extracting oil, starch and other edible components, the physiochemical and functional characteristics as well as their applications in food industry. Different extraction methods can affect functional and nutritional properties to a certain extent. At present, mechanical compressing, alkaline method and alkali extraction-acid precipitation are the most suitable methods for production of its oil, starch and protein in food industry, respectively. Based on traditional extraction methods, combination of innovative techniques aimed at yield and physiochemical characteristics is essential for the comprehensive utilization of nutrients. In addition, tiger nut has the radical scavenging ability, invitro inhibition of lipid peroxidation, anti-inflammatory and anti-apoptotic effects, displays medical properties. It has been made to milk, snacks, beverages and gluten-free bread. Despite their ancient use for food and feed and the many years of intense research, tiger nut and their components still deserve further exploitation on the functional properties, modifications and intensive processing to make them suitable for industrial production.

Point 3: The entire format is not according to the journal style. Authors must need to read journal guidelines and follow the journal format.

Response 3: Thanks for your comment. We are glad to accept your suggestion. We have revised the entire format according to the journal style. The revised parts are shown in the text in red.

Point 4: Figure 1. increase the font size of descriptors

Response 4: Thanks for your comment. We have made the revisions.

Page 2, line 53: Figure displayed in attachment.

Point 5: Section 4.2. Composition and Nutritional Evaluation of Tiger Nut Protein: Please provide detailed discussion

Response 5: Thanks for your comment. We have made the revisions.

Page 11, line 371:

Plant proteins can now be regarded as functional ingredients or as biologically active components more than as essential nutrients. Because functional properties of protein often have a tight relationship with its amino acid profile, so the amino acid composition represents the potential quality of a plant protein. Researchers evaluated the nutritional value of tiger nut protein by considering both essential amino acid profile and protein digestibility. Tiger nut protein contains 18 kinds of amino acid, of which the essential amino acids accounts for 46.03%, largely exceeding the value specified in the WHO/FAO model (36%), higher than soy protein (41.3%), and close to egg protein (48.8%). According to the literature, lysine is the first limiting amino acid of some nuts (Brazil nuts, macadamia nuts and almonds and some cereal grains (rice, white flour, corn, etc.). But the amounts of lysine in tiger nut are much greater, accounting for 15.4% of essential amino acids, which is higher than 13.3% of soybean [62]. Therefore, with respect to lysine amino acid contents, cereal and tiger nut proteins are nutritionally complementary. The first limiting amino acids of the tiger nut protein is methionine. Taking the score of the first limiting amino acid as the protein amino acid score, the amino acid score of tiger nut protein is 78.9, higher than soy protein (51.4), and lower than egg protein (97.2). The digestibility of the tiger nut protein was also evaluated. Results showed that tiger nut protein has 76% of in vitro digestibility, slightly lower than soy protein (86%) [62]. In conclusion, the protein derived from tiger nut is considered high quality based on amino acid profile and bioavailability. On the other hand, the protein of tiger nut is limited in methionine. We recommend the tiger nut as the sources of complementary protein.

Point 6: Section 4.3. Activity of Tiger Nut Protein, which activity?

Response 6: Thanks for your comment. We have made the revisions.

Page 11, line 392: “Physiochemical characteristics of Tiger Nut Protein”.

Point 7: 5.2. Comprehensive Utilization of Tiger Nut By-Products. Please provide in Table form for the tiger nut by-products, source and food product

Response 7: Thanks for your comment. We are glad to accept your suggestion.

Page 14, line 476:

Table 5. Application of tiger nut by-products milk in the food industry

Tiger Nut By-Products

Source

Food Product

liquid phase

phenolic compounds

natural antioxidants

replacing water

cooked pork liver meat product

carbon source for growth of probiotic bacteria

fermentable substrates

solid phase

Tiger nut fiber

cereal foods (like chips, breakfast cereal or dry pasta)

pork pies, pork burgers and dry-cured sausages

Point 8: 6. Medicinal Properties of Tiger Nut should be more elaborated. Is it a whole tiger nut or including by-products of it? Detailed discussion should be necessary

Response 8: Thanks for your comment. We have made the revisions.

Page 15, line 547:

Tiger nut have been shown to be good sources of bioactive substances. It contains polyphenols, flavones, minerals, essential fatty acid, and vitamins C, D and E, etc. And thus, it might possess medicinal properties. Ademosun & Oboh sought to investigate the radical scavenging ability and invitro inhibition of lipid peroxidation, α-amylase, and α-glucosidase activities of the tiger nut extracts. The antioxidants of aqueous extracts of the tiger nut as typified by1,1-diphenyl-2 picrylhydrazyl (DPPH) and hydroxyl (OH) radicals scavenging abilities and inhibition of Fe2+-induced malondialdehyde (MDA) production in rat’s pancreas in vitro were assessed. Results showed that the EC50 of DPPH· and OH· scavenging abilities, Fe2+-chelating ability, inhibition of Fe2+-induced MDA production, and inhibition of α-amylase and α-glucosidase activities by aqueous extracts of the tiger nut are 9.63 ± 0.7 mg/mL, 3.01 ± 0.12 mg/mL, 0.72 ± 0.07 mg/mL, 2.09 ± 0.10 mg/mL, and 0.76 ± 0.06 mg/mL, respectively [92]. The findings support the hypothesis that tiger nut maybe beneficial in the management of type 2 diabetes. In other studies, the tiger nut powder was administered to the rats daily showed anti-inflammatory and anti-apoptotic effects to prevent testicular dysfunction [93], ameliorated male arousal [94], reduced diarrheal symptoms in albino rats [95] and reduced oxidative stress in liver and inflammatory with atherosclerosis [96]. Results above are most likely related to the presence of alkaloids, quercetin, vitamins, steroids, and zinc etc. in tiger nut. Therefore, in addition to be the food and industrial materials, the tiger nuts may also be developed into functional foods. Nevertheless, future investigations are warranted to confirm these effects of tiger nut in humans and to identify the specific ingredients that play the medicinal role.

Point 9: References: none of the references are according to journal format. It seems possible that the authors are no aware of Foods guidelines

Response 9: Thanks for your comment. We have made the revisions of references according to journal format. The revised parts are shown in the text in red.

Point 10: Scientific names should be in Italics and journal names should be in Italics (according to food references style). Authors should check all the references and make sure to follow journal guidelines.

Response 10: Thanks for your comment. We have made the revisions of references according to journal format. The revised parts are shown in the text in red.

Point 11: Find the attached document for other suggestions

Response 11: Thank you very much for your comments. We have made the revisions of the suggestions in the attached document. The revised parts are shown in the text in red.

Reviewer 2 Report

This review on tiger nuts is really expansive, interesting and well written. It was very interesting to read. I only have some minor comments to be addressed:

  • P2 L45 - Figure 1 caption: There is no need to capitalize first letters in each word in the figure caption.
  • I would suggest the authors to re-think the use of the word "tiger nut" throughout the manuscript. There is definitely some parts with excesive use of the word, where there is no need to use it, since it is clear that the text is about tiger nuts, and not some other tuber.
  • There are missing spaces between words and the brackets containing references in the text. Please check the entire manuscript and add spaces where they are missing.
  • P4, L125-126: Which oil extraction process is used on the industrial scale? You mentioned that the combination of extraction techniques cannot be used because they are not feasible, but you did not mention which one is currently used in the industry. Please list some examples of the processes which are used on the industrial scale for the production of tiger nut oil.
  • P4,Table 2. Please add SD and statistical significance data it the table.
  • P6, Table 3. Please add statistical significance data it the table.
  • P7, Figure 2 caption: There is no need to capitalize first letters in each word in the figure caption.
  • P8, L225-227: In the sentence "The study showed..." there is no reference to a study. Please add the reference.
  • P10, L360: "The researched and application of tiger nut protein have been studied relative...". I think there is something written wrong in this sentence. Please correct.
  • P11, Figure 3 caption: There is no need to capitalize first letters in each word in the figure caption.
  • P12, L412-413: "The results indicated that it have a high potential for probiotic microorganisms’..." Please correct have to has.
  • P12, L417: There is an "r" missing in "tige nut milk"
  • References are not formatted according to the Instructions for Authors.

Author Response

We want to express our sincere gratitude to the reviewer for your effort and patience in reviewing our manuscript. We appreciate for Reviewer’s warm work earnestly. We addressed all the points as summarized below. All the changes were highlighted in red.

Point 1: P2 L45 - Figure 1 caption: There is no need to capitalize first letters in each word in the figure caption.

Response 1: Thanks for your comment.

Page 2, line 54:

We changed the caption of the figure in the article to: Figure 1. The main nutrition composition and nutrient-based applications of tiger nut.

Point 2: I would suggest the authors to re-think the use of the word "tiger nut" throughout the manuscript. There are definitely some parts with excessive use of the word, where there is no need to use it, since it is clear that the text is about tiger nuts, and not some other tuber

Response 2: Thanks for your comment. We have made the revisions for the use of the word “tiger nut”. The revised parts are shown in the text in red.

Point 3: There are missing spaces between words and the brackets containing references in the text. Please check the entire manuscript and add spaces where they are missing.

Response 3: Thanks for your comment. We have checked the entire manuscript and add spaces where they are missing.

Point 4: P4, L125-126: Which oil extraction process is used on the industrial scale? You mentioned that the combination of extraction techniques cannot be used because they are not feasible, but you did not mention which one is currently used in the industry. Please list some examples of the processes which are used on the industrial scale for the production of tiger nut oil.

Response 4: Thanks for your comment. We have made the revisions.

Page 4, line 148:

The SE has higher oil yield but easy to produce solvent residues. Tiger nut oil is produced by cold pressed in food industry, which has better quality, but the oil yield is relatively lower. Due to complexity and high equipment cost, unconventional oil extraction methods are difficult to be applied to industrial production.

Point 5: P4, Table 2. Please add SD and statistical significance data it the table.

Response 5: Thanks for your comment. SD and statistical significance data have been added in the table.

Page 5, line 168:

Table 2. Fatty acid composition of tiger nut oil under different extraction methods

Fatty acid

SE

ME

MAE

MUAAEE

SC-CO2

SBE

Palmitic acid (C16:0)

11.65 ± 0.28c

11.98 ± 0.00b

12.08 ± 0.16ab

11.86 ± 0.23bc

12.36 ± 0.01a

12.14 ± 0.01ab

Palmitoletic Acid (C16:1)

0.26 ± 0.00a

-*

0.22 ± 0.00b

0.25 ± 0.00a

- *

- *

Cydonic acid (C17:0)

0.07 ± 0.00a

- *

0.06 ± 0.00a

0.07 ± 0.00a

- *

- *

Stearic acid (C18:0)

2.43 ± 0.05d

4.92 ± 0.00a

2.24 ± 0.04e

2.37 ± 0.07d

4.76 ± 0.00c

4.84 ± 0.00b

Oleic acid (C18:1)

74.52 ± 0.51bc

73.97 ± 0.01c

75.60 ± 0.58a

74.73 ± 0.46b

73.83 ± 0.01c

74.10 ± 0.01bc

Linoleic acid (C18:2)

9.63 ± 0.12a

8.75 ± 0.00b

8.85 ± 0.13b

9.46 ± 0.18a

8.86 ± 0.00b

8.75 ± 0.00b

α-linolenic acid (C18:3)

0.21 ± 0.00c

0.38 ± 0.00b

0.91 ± 0.00a

0.2 ±

0.00d

0.19 ± 0.00e

0.17 ± 0.00f

Arachidic acid (C20:0)

0.45 ± 0.00a

- *

0.41 ± 0.00c

0.42 ± 0.00b

- *

- *

Eicosenoic acid (C20:1)

0.28 ± 0.00a

-*

0.28 ± 0.00a

0.29 ± 0.00a

- *

- *

Saturated fatty acids

14.25 ± 0.05e

16.90 ± 0.01c

14.69 ± 0.05d

14.65 ± 0.07d

17.12 ± 0.01a

16.98 ± 0.01b

Monounsaturated fatty acids

75.06 ± 0.12c

73.97 ± 0.01de

76.16 ± 0.12a

75.24 ± 0.13b

73.83 ± 0.01e

74.10 ± 0.01d

Polyunsaturated fatty acids

9.84 ± 0.07a

9.14 ± 0.00c

9.04 ± 0.07c

9.72 ± 0.09b

9.05 ± 0.00c

8.92 ± 0.00d

Reference

[19]

[32]

[23]

[19]

[32]

[32]

* Not detected

Different letters in a row indicate significant differences at the 5% level.

SE: Soxhlet extraction; ME: mechanical expressing; GAME: gas assisted mechanical expression; MEEA: mechanical compressing with enzyme assistance; MAE: microwave-assisted extraction; MUAAEE: microwave-ultrasonic assisted aqueous enzymatic extraction; SC-CO2: supercritical carbon dioxide fluid extraction; SBE: subcritical n-butane extraction.

Point 6: P6, Table 3. Please add statistical significance data it the table.

Response 6: Thanks for your comment. The statistical significance data have been added in the table.

Page 6, line 214:

Table 3. Physicochemical characteristics of tiger nut oil under different extraction methods

Method

Refractive Index (25)

Acid Value (mg/g)

Peroxide Value (meqO2/kg)

Saponification (mg/g)

Iodine Value (g/100g)

Reference

SE

1.46±0.00a

4.15±0.32a

16.26±0.53b

187.25±1.42a

84.78±1.13ab

[19]

ME

1.48±0.00b

1.90±0.01e

15.76±0.00b

174.53±0.62b

67.35±0.49c

[32]

MAE

1.46±0.00b

2.26±0.17d

8.78±0.42c

185.67±1.37a

85.41±1.06a

[23]

MUAAEE

1.46±0.00b

2.35±0.13d

7.63±0.35d

187.52±1.23a

83.67±0.85b

[19]

SC-CO2

1.46±0.00b

3.39±0.16b

15.76±0.00b

175.33±1.61b

65.60±0.14d

[32]

SBE

1.46±0.00b

2.86±0.02c

23.64±0.00a

176.71±0.81b

66.15±0.64cd

[32]

Different letters in a row indicate significant differences at the 5% level.

SE: Soxhlet extraction; ME: mechanical expressing; GAME: gas assisted mechanical expression; MEEA: mechanical compressing with enzyme assistance; MAE: microwave-assisted extraction; MUAAEE: microwave-ultrasonic assisted aqueous enzymatic extraction; SC-CO2: supercritical carbon dioxide fluid extraction; SBE: subcritical n-butane extraction.

Point 7: P7, Figure 2 caption: There is no need to capitalize first letters in each word in the figure caption.

Response 7: Thanks for your comment.

Page 8, line 255:

We changed the caption of the figure 2 to: Figure 2. The preparation of tiger nut oil in food industry and by-products generated.

Point 8: P8, L225-227: In the sentence "The study showed..." there is no reference to a study. Please add the reference.

Response 8: Thanks for your comment. We have made the revisions.

Page 9, line 292:

The study showed wide variations in some physicochemical properties of starches isolated from the two types of tubers (black and yellow) [54].

P.T. Akonor, C. Tortoe, C. Oduro-Yeboah, E.A. Saka, J. Ewool. Physicochemical, microstructural, and rheological characterization of tigernut (Cyperus esculentus) Starch. International Journal of Food Science 2019, 2019, 1-7.

Point 9: P10, L360: "The researched and application of tiger nut protein have been studied relative...". I think there is something written wrong in this sentence. Please correct.

Response 9: Thanks for your comment. We have made the revisions.

Page 12, line 420:

"There are relatively few researches and applications on tiger nut protein, the future development should focus on the following technologies..."

Point 10: P11, Figure 3 caption: There is no need to capitalize first letters in each word in the figure caption.

Response 10: Thanks for your comment.

Page 13, line 445:

We changed the caption of the figure 3 to: "Figure 3. Applications of whole tuber of tiger nut in food industry".

Point 11: P12, L412-413: "The results indicated that it have a high potential for probiotic microorganisms’..." Please correct have to has.

Response 11: Thanks for your comment. We have made the revisions.

Page 14, line 494:

The results indicated that it has a high potential for probiotic microorganisms’...

Point 12: P12, L417: There is an "r" missing in "tige nut milk"

Response 12: Thanks for your comment. We have made the revisions.

Page 14, line 497:

The solid by-products of tiger nut milk are the main source of fiber.

Point 13: References are not formatted according to the Instructions for Authors.

Response 13: Thanks for your comment. We have made the revisions of references according to journal format. The revised parts are shown in the text in red.

Reviewer 3 Report

In this manuscript, the authors summarised the literature on the nutritional, processing, function and applications of Cyperus esculentus, also known as tiger nut. Overall the paper is easy to read and understand.

In terms of scientific content, it seems that there is a major overlapping in the scope of this review with those that were published previously, and even in recent years. Hence, it is crucial for the authors to acknowledge that there are existing reviews on this species in the introduction section and further state or highlight the novelty of their work - which is unclear in the present manuscript.

The manuscript is divided according to the chemical components of the tiger nut, including its oil, starch and protein before elaborating on the products derived from the tiger nut. Such organisation seems interesting enough, for chemists, but it will be complete if there is a mention of the small molecules or secondary metabolites that are mostly responsible for the bioactivities.

Secondly, perhaps it will be of interest for the authors to include their perspectives on the various chemical components of tiger nut. One may be curious to know, based on our scientific knowledge, which component has higher potential to be commercialised?

The other thing would be how the tiger nut components fare when it comes to various nutritional aspects, in comparison with other members of the genus or related genus or nuts in general. This is important considering the interest in turning these into functional products.

The authors compiled the results of previous researchers and make comparison of the values obtained. This can be observed in the tables. However, one need to take into account the differences in the methodology adopted by different researchers render it almost impossible to make a direct comparison based on the values obtained, for instance in Table 4.

A separate section on further perspectives based on the authors' knowledge and opinion would certainly enhance the value of this review.

Other comments:

  1. Suggest to include photos of this species and some of the products.
  2. Lines 46-47: please elaborate on how these factors may affect its nutritional content.
  3. Lines 128-129: please explain with examples
  4. Lines 197-198: please explain (similar to #3)
  5. Line 237: the similarities can be stated here
  6. Lines 272-295: any comparison with starch from other sources?
  7. Lines 325-326: please provide a reference
  8. Lines 466-473: this section is too brief, it should be expanded

Author Response

We want to express our sincere gratitude to the reviewer for your effort and patience in reviewing our manuscript. We appreciate for Reviewer’s warm work earnestly. We addressed all the points as summarized below. All the changes were highlighted in red.

Point 1: In terms of scientific content, it seems that there is a major overlapping in the scope of this review with those that were published previously, and even in recent years. Hence, it is crucial for the authors to acknowledge that there are existing reviews on this species in the introduction section and further state or highlight the novelty of their work - which is unclear in the present manuscript.

Response 1: Thanks for your comment. We are glad to accept your suggestion. And the introduction section has been revised.

Page 2, line 64:

The selection of appropriate methods are key steps in comprehensive utilization of it. Although tiger nut is an ancient crop, people have made some contributions in its component analysis, physicochemical properties and product development, but the breadth and depth research are far less than that of soybean, peanut and other nut oil crops. The purpose of this article is to make this crop of interest to scientists for further research by outlining its current state of research and production. At present, the reviews of tiger nut mainly focus on recovered oil and milk processing, based on the extraction method and physicochemical properties, the analysis of nutrients is of great significance for the comprehensive utilization of tiger nut. This review article summarizes the current knowledge about the major nutrients of the tiger nut, the processing methods, their physicochemical properties, functional characteristics and the applications in food industry.

Point 2: The manuscript is divided according to the chemical components of the tiger nut, including its oil, starch and protein before elaborating on the products derived from the tiger nut. Such organisation seems interesting enough, for chemists, but it will be complete if there is a mention of the small molecules or secondary metabolites that are mostly responsible for the bioactivities.

Response 2: Thanks for your comment. We have made the revisions.

Page 12, line 423:

Besides the macromolecular nutrients such as oil, starch and protein, studies have shown that there are small molecules or secondary metabolites that are mainly responsible for biological activity, including flavonoids, minerals, vitamins and Stigmasterol [17]. Among all of the secondary metabolites, the flavonoids quercetin and myricetin show a wide range of biological activities that include strong anti-oxidant, anticarcinogenic, anti-inflammatory and antidiabetic effects. Due to the presence of quercetin, this tuber has aphrodisiac activity, enhancing male sexual libido and performance [17]. Minerals in tiger nut are sodium, potassium, calcium, iron, magnesium, zinc, copper and phosphorus. The high potassium (110.70 ̵ 121.95 mg/100 g) to low sodium ration (99.95 ̵ 105.6 mg/100 g) of tiger nuts may be imperative in diet formulations for patients with high blood pressure and edema [5]. Moreover, the presence of vitamin C and E makes the tuber has the function of preventing scurvy and promoting liver detoxification [17]. The cholesterol-lowering activity and the protective effect on cardiovascular disease of tuber may be due to the stigmasterol. However, the isolation, identification and quantification of these active ingredients, which have not been adequately studied, are necessary for the development of functional properties of tiger nut.

Point 3: Secondly, perhaps it will be of interest for the authors to include their perspectives on the various chemical components of tiger nut. One may be curious to know, based on our scientific knowledge, which component has higher potential to be commercialised?

Response 3: Thanks for your comment. We have made the revisions.

Page 7, line 247:

From the point of view of commercial production, oily bean has high oil yield and oil quality comparable to that of olive oil, and has the potential for industrialized mass production.

Page 10, line 308:

Tiger nut starch has similar viscosity characteristics to native starch, but its gel texture properties (hardness, elasticity, cohesion and chewiness) are higher than those of traditional corn and sweet potato starch. In addition, its freeze-thaw stability is better than that of corn starch, which can be used in cold drinks and frozen foods, and has a wide range of application values. Therefore, it has potential commercial value and has been used in food and pharmaceutical industry.

Page 11, line 371:

Plant proteins can now be regarded as functional ingredients or as biologically active components more than as essential nutrients. Because functional properties of protein often have a tight relationship with its amino acid profile, so the amino acid composition represents the potential quality of a plant protein. Researchers evaluated the nutritional value of tiger nut protein by considering both essential amino acid profile and protein digestibility. Tiger nut protein contains 18 kinds of amino acid, of which the essential amino acids accounts for 46.03%, largely exceeding the value specified in the WHO/FAO model (36%), higher than soy protein (41.3%), and close to egg protein (48.8%). According to the literature, lysine is the first limiting amino acid of some nuts (Brazil nuts, macadamia nuts and almonds and some cereal grains (rice, white flour, corn, etc.). But the amounts of lysine in tiger nut are much greater, accounting for 15.4% of essential amino acids, which is higher than 13.3% of soybean [62]. Therefore, with respect to lysine amino acid contents, cereal and tiger nut proteins are nutritionally complementary. The first limiting amino acids of the tiger nut protein is methionine. Taking the score of the first limiting amino acid as the protein amino acid score, the amino acid score of tiger nut protein is 78.9, higher than soy protein (51.4), and lower than egg protein (97.2). The digestibility of the tiger nut protein was also evaluated. Results showed that tiger nut protein has 76% of in vitro digestibility, slightly lower than soy protein (86%) [62]. In conclusion, the protein derived from tiger nut is considered high quality and commercial value based on amino acid profile and bioavailability. On the other hand, the protein of tiger nut is limited in methionine. We recommend the tiger nut as the sources of complementary protein.

Point 4: The other thing would be how the tiger nut components fare when it comes to various nutritional aspects, in comparison with other members of the genus or related genus or nuts in general. This is important considering the interest in turning these into functional products.

Response 4: Thanks for your comment. We have made the revisions.

Page 7, line 230:

In the process of oil extraction, a part of polyphenols will be extracted, which makes tiger nut oil with antioxidant properties. Moreover, high monounsaturated fatty acid (oleic acid) gives it the effect of improving blood circulation and regulating physiological metabolism.

Page 11, line 367:

The various components of tiger nut protein have different effects on their functional properties, it is meaningful to find protein components suitable for the production of different products for the comprehensive utilization of tiger nut protein.

Page 14, line 497:

Compared with other sources of dietary fiber (oat bran, rice bran, sugarcane waste), Tiger nut fiber has higher water holding capacity, oil holding capacity and emulsification stability, and low water absorption [78]. Studies have shown that foods rich in dietary fiber are essential for preventing colon cancer, constipation, obesity, cardiovascular disease [9].

Point 5: The authors compiled the results of previous researchers and make comparison of the values obtained. This can be observed in the tables. However, one need to take into account the differences in the methodology adopted by different researchers render it almost impossible to make a direct comparison based on the values obtained, for instance in Table 4.

Response 5: Thanks for your comment. We have made the revisions.

Page 8, line 277:

Table 4. Extraction technologies of tiger nut starch and protein

Method

Condition

Amylose

(g/g)

Amylopectin

(g/g)

Protein Content

Reference

Alkaline method

solid-liquid ratio of 1:15, 0.15% sodium hydroxide, soak 30-40 min, 45℃ drying 24h

16.18

83.82

-

[51]

sodium metabisulfite extraction

sodium metabisulfite solution (0.075% w/v) dispersing, 150 µm muslin cloth filtering, air dried for 10 h, further dried for 5 h at 50°C

11.5

88.5

-

[53]

water extraction

milled with water, Gauze filter, 150 μ mesh filter, 40°C drying 18 h

19.1

80.9

-

[54]

alkali extraction-acid precipitation

alkali extraction pH 8.5, extraction time 1.5 h, acid precipitation pH 4.5, incubated at 4 °C for 2 h

-

-

72%

[58]

ammonium precipitation

pressing of the pre-soaked mush, filter through a 13-µm pore-size membrane, 50 % w/v ammonium sulphate, 16 h, 6- to 8-kDa cut-off membrane tube dialyzing

-

-

83.5%

[59]

AOT/isooctane reverse micelle extraction

The pre-extraction condition are as follows: 0.05 g/mL feed volume, 40°C, 86 min, pH 7, the AOT mass concentration is 0.12 g/mL, the water content is 16, the KCl concentration is 0.02 mol/L. The back-extraction conditions were as follows: stripping time was 5 min, the pH of the aqueous phase was 11, and the KCl concentration was 1.2 mol/L.

-

-

-

[60]

The content of amylose and amylopectin was determined by colorimetric method based on amylase-iodine complex formation. The respective protein content in table was determined using the Kjeldahl method (N × 6.25).

Point 6: A separate section on further perspectives based on the authors' knowledge and opinion would certainly enhance the value of this review.

Response 6: Thanks for your comment. We have made the revisions

Page 16, line 585:

Tiger nut is an excellent and healthy crop from both a nutritional and commercial point of view. Green production and comprehensive utilization are the future development direction. In the process of full utilization of tiger nut, the first step is to extract oil, and the extraction method of oil has a certain influence on the yield and physiochemical properties of subsequent starch, protein and fiber. Therefore, the use of innovative combination methods to fractionate and extract the nutrients oil, starch, protein and fiber in tuber is worth thinking about in the future. It is meaningful to make modification of natural tiger nut starch and develop new varieties of starch, makes it adaptable to different processing conditions and use it in biological preservative films. In addition, investigating the functional properties of protein, and using it in baking and nutritional meal replacement foods is crucial to the study of tiger nut protein. Due to its high fiber content, it can be used in the development of food for improving gut health and weight loss.

Point 7: Suggest to include photos of this species and some of the products.

Response 7: Thanks for your comment. We have made the revisions. We have added a picture of this species in the text.

Page 13, line 446: Figure displayed in attachment.

Figure 4. Tiger nut tuber and its products (a. tuber; b. edible oil; c. liquor; d. crunchy bio toast; e. solid powder drink; f. meal replacement powder)

Point 8: Lines 46-47: please elaborate on how these factors may affect its nutritional content.

Response 8: Thanks for your comment. We have made the revisions.

Page 2, line 55:

The nutritional value of tiger nut and its products are dependent on varieties, soil conditions, growth environment, cultivation techniques and storage conditions [5, 11-13]. Nina compared the nutritional content of black, brown and yellow oil bean varieties and found that the crude fiber of the black cultivar was higher than the brown and yellow varieties and protein content of the brown cultivar was higher than that of the black and yellow cultivar [5]. In addition, combined application of wood biochar and cow dung biochar with green manure improved soil fertility and increased growth and yield of tiger nut [11]. Tuber varied considerably among the genotypes. Refrigeration of tiger nuts resulted in higher activities of α-amylase and lipase, whilst ambient or elevated storage showed higher sugar content of tiger nuts [13].

Point 9: Lines 128-129: please explain with examples

Response 9: Thanks for your comment. We have made the revisions.

Page 4, line 150:

Since a single extraction technique usually has certain shortcomings, combination of conventional with novel techniques is the direction of future development. It has been confirmed that the oil yield by GAME (28.48 g/100 g) is higher than that of single ME (19.94 g/100 g) and SC-CO2 (27.79 g/100 g).

Point 10: Lines 197-198: please explain (similar to #3)

Response 10: Thanks for your comment. We have made the revisions.

Page 7, line 222:

Microwave and ultrasound assistance can promote the release of active substances such as phenols and improve the oxidation stability of oil, which shows low peroxide value indicated in MAE (8.78 ± 0.42 meqO2/kg) and MUAAEE (7.63 ± 0.35 meqO2/kg). On the basis of traditional extraction, combination with other methods for assistance to obtain high-quality and yield oil may be the future development direction.

Point 11: Line 237: the similarities can be stated here

Response 11: Thanks for your comment. We have made the revisions.

Page 8, line 270:

The Tiger nut starch has similar characteristics as corn starch, potato starch and sweet potato starch, such as oval particles shape, hygroscopicity and adhesiveness [48, 49].

Point 12: Lines 272-295: any comparison with starch from other sources?

Response 12: Thanks for your comment. We have made the revisions.

Page 10, line 323:

Indeed, this starch showed excellent binder activity, mechanical strength, fluidity and release properties when it was used as glidant and binder in the production of metronidazole and ascorbic acid tablets [52, 56]. The native tiger nut starch showed flow properties comparable to maize and potato starch as well as excellent binder activity when metronidazole tablets’ hardness and friability were evaluated [48].

Point 13: Lines 325-326: please provide a reference

Response 13: Thanks for your comment. We have made the revisions.

Page 11, line 366:

The reason for this difference was possibly because that the proximate composition of tiger nut was mainly dependable on their geographical origin [12].

P.A. Asare, R. Kpankpari, M.O. Adu, E. Afutu, A.S. Adewumi. Phenotypic Characterization of Tiger Nuts (Cyperus esculentus L.) from Major Growing Areas in Ghana. Scientific World Journal 2020, 1, 7232591.

Point 14: Lines 466-473: this section is too brief, it should be expanded

Response 14: Thanks for your comment. We have made the revisions.

Page 15, line 546:

Tiger nut have been shown to be good sources of bioactive substances. It contains polyphenols, flavones, minerals, essential fatty acid, and vitamins C, D and E, etc. And thus, it might possess medicinal properties. Ademosun & Oboh sought to investigate the radical scavenging ability and invitro inhibition of lipid peroxidation, α-amylase, and α-glucosidase activities of the tiger nut extracts. The antioxidants of aqueous extracts of the tiger nut as typified by1,1-diphenyl-2 picrylhydrazyl (DPPH) and hydroxyl (OH) radicals scavenging abilities and inhibition of Fe2+-induced malondialdehyde (MDA) production in rat’s pancreas in vitro were assessed. Results showed that the EC50 of DPPH· and OH· scavenging abilities, Fe2+-chelating ability, inhibition of Fe2+-induced MDA production, and inhibition of α-amylase and α-glucosidase activities by aqueous extracts of the tiger nut are 9.63 ± 0.7 mg/mL, 3.01 ± 0.12 mg/mL, 0.72 ± 0.07 mg/mL, 2.09 ± 0.10 mg/mL, and 0.76 ± 0.06 mg/mL, respectively [92]. The findings support the hypothesis that tiger nut maybe beneficial in the management of type 2 diabetes. In other studies, the tiger nut powder was administered to the rats daily showed anti-inflammatory and anti-apoptotic effects to prevent testicular dysfunction [93], ameliorated male arousal [94], reduced diarrheal symptoms in albino rats [95] and reduced oxidative stress in liver and inflammatory with atherosclerosis [96]. Results above are most likely related to the presence of alkaloids, quercetin, vitamins, steroids, and zinc etc. in tiger nut. Therefore, in addition to be the food and industrial materials, the tiger nuts may also be developed into functional foods. Nevertheless, future investigations are warranted to confirm these effects of tiger nut in humans and to identify the specific ingredients that play the medicinal role.

Reviewer 4 Report

Dear authors,

This review articles addresses the potential values for tiger nuts in the oil industry as well as their by-products. The article is well-structured. The figures needs to be in a better resolution. Their are several corrections in the attached file, so please follow. 

Besides, the article lacks any information regarding the global total production of tiger nuts or the quantities of the produced oil. Hence, checking this information through FAO statistics is highly recommended.

Author Response

We want to express our sincere gratitude to the reviewer for your effort and patience in reviewing our manuscript. We appreciate for Reviewer’s warm work earnestly. We addressed all the points as summarized below:

Point 1: This review articles addresses the potential values for tiger nuts in the oil industry as well as their by-products. The article is well-structured. The figures needs to be in a better resolution. Their are several corrections in the attached file, so please follow. 

Response 1: Thank you very much for your comments. We have made the revisions of the suggestions in the attached document. The revised parts are shown in the text in red.

Point 2: Besides, the article lacks any information regarding the global total production of tiger nuts or the quantities of the produced oil. Hence, checking this information through FAO statistics is highly recommended.

Response 2: Thanks for your comment. We have made the revisions.

Page 1, line 32:

Moreover, it is an important representative crop of the Spanish Mediterranean region, with an annual production of 9000 metric tons [2].

Round 2

Reviewer 1 Report

Authors answered the questions/suggestions raised by me. However, scientific name in the title should be in Italics before final acceptance. 

Reviewer 3 Report

The authors have revised the manuscript according to the reviewers' comments. I do not have further comments but to suggest to look through the manuscripts for typographical errors, and to ensure that all abbreviations are spelt in full when they appear for the first time.